# IMPROVING LARGE LANGUAGE MODEL FINE-TUNING FOR SOLVING MATH PROBLEMS

## ABSTRACT

Despite their success in many natural language tasks, solving math problems remains a significant challenge for large language models (LLMs). A large gap exists between LLMs' pass-at-one and pass-at-N performance in solving math problems, suggesting LLMs might be close to finding correct solutions, motivating our exploration of fine-tuning methods to unlock LLMs' performance. Using the challenging MATH dataset, we investigate three fine-tuning strategies: (1) solution fine-tuning, where we fine-tune to generate a detailed solution for a given math problem; (2) solution-cluster re-ranking, where the LLM is fine-tuned as a solution verifier/evaluator to choose among generated candidate solution clusters; (3) multi-task sequential fine-tuning, which integrates both solution generation and evaluation tasks together efficiently to enhance the LLM performance. With these methods, we present a thorough empirical study on a series of PaLM 2 models and find: (1) The quality and style of the step-by-step solutions used for fine-tuning can make a significant impact on the model performance; (2) While solution re-ranking and majority voting are both effective for improving the model performance when used separately, they can also be used together for an even greater performance boost; (3) Multi-task fine-tuning that *sequentially* separates the solution generation and evaluation tasks can offer improved performance compared with the solution fine-tuning baseline. Guided by these insights, we design a fine-tuning recipe that yields approximately 58.8% accuracy on the MATH dataset with fine-tuned PaLM 2-L models, an 11.2% accuracy improvement over the few-shot performance of pre-trained PaLM 2-L model with majority voting.

## 1 INTRODUCTION

Solving mathematical problems is a challenging task for even the state-of-the-art large language models (LLMs), e.g., GPT-4 (OpenAI, 2023) and PaLM 2 (Anil et al., 2023), since it requires the ability of creative thinking, mathematical reasoning and numerical calculation. However, LLMs are already showing potential of achieving better performance on this math problem solving task, as the likelihood of LLM's being able to find a correct answer is significantly higher when they are allowed to attempt the problem several times. For example, with greedy decoding the pre-trained PaLM 2-L can achieve around 33.4% accuracy, but there is at least one correct solution out of 64 sampled solutions (pass@64) 79.4% of the time (Table 2) when using temperature sampling. This large performance gap suggests that LLMs can be capable of generating correct solutions while struggling to discriminate correct from incorrect solutions.

Therefore, we study task-specific fine-tuning methods that can improve the LLM's solution generation and evaluation ability such that the aforementioned performance gap can be reduced. Specifically, we explore three fine-tuning methods:

(1) **Supervised step-by-step solution fine-tuning (SSFT)**. As a baseline method, we investigate whether the pre-trained LLMs can benefit from a supervised fine-tuning stage. To this end, we fine-tune the LLMs to generate the step-by-step solution and final answer as in Lightman et al. (2023).

(2) **Solution-cluster Re-ranking (SCR)**. To enhance the LLM's solution evaluation ability, we continue fine-tuning the generator as a solution evaluator for candidate solution re-ranking. While such a solution sample-rank, or re-ranking, has been investigated in previous work (Cobbe et al., 2021), we propose a new technique that can bring the benefits of both majority voting (Wang et al., 2023)

and re-ranking together, while reducing ranking costs. Specifically, we first group the candidate answers into different clusters according to their mathematical equivalence, which is an intermediate step in majority voting. Then, we apply the solution evaluator to the solutions in the most-frequent clusters to gain further improvement over the majority voting results.

(3) **Multi-task Sequential Fine-tuning**. Apart from the solution evaluation task, we are also interested in improving the LLM's performance on the solution generation task and exploring whether the training objective of solution evaluation can be beneficial to the solution generation model. To this end, we propose a *sequential* multi-task learning setting for the generation model where the solution evaluation task is formatted in the form of a natural language generation task, such that its training objective can provide meaningful supervision signal to the solution generation model. Concretely, we fine-tune the model in a sequential manner: fine-tuning (1) as a solution generator (SSFT), (2) as a solution evaluator (SCR), (3) as a generator (SSFT) again.

We conduct comprehensive experiments on the challenging MATH (Hendrycks et al., 2021b) dataset with PaLM 2-S* and PaLM 2-L– the small and large variants of PaLM 2 respectively (Anil et al., 2023) – which leads to these findings:

- For SSFT, the quality and the style of the step-by-step solutions can make a large impact on fine-tuned model, as they benefit more from fine-grained, well-formatted solutions.

- Re-ranking solutions in the most-frequent solution clusters can yield better performance than re-ranking all the solutions while simultaneously achieving better computational efficiency, which we believe can be a better standard practice for future work.

- Our proposed multi-task sequential fine-tuning can more effectively improve the solution generation model performance compared with supervised solution fine-tuning only, showing the benefit of training the model for both solution generation and evaluation tasks, presenting a successful attempt of leveraging learning signal of a binary evaluation task for a generation model.

## 2 BACKGROUND

Mathematical problem solving is an important task (Hendrycks et al., 2021a;b; Cobbe et al., 2021) for measuring the LLMs' reasoning and numerical computation abilities. In this work, we focus on the MATH dataset (Hendrycks et al., 2021b), which consists of problems collected from high school math competitions, along with the human-written solutions containing both natural language explanations and the final ground-truth solutions. MATH dataset is challenging to even the recent start-of-the-art large language models (LLMs), such as GPT-4 (OpenAI, 2023) and PaLM 2 (Anil et al., 2023), since they can only achieve 42.5% and 33.2% pass-at-1 accuracy (Anil et al., 2023). The accuracy is usually calculated through an automatic grading function $g$ checking the mathematical equivalence between the ground truth solution $A$ and the model solution $\tilde{A}$:

$$g(A, \tilde{A}) = \begin{cases} 1 & \text{if } \tilde{A} \text{ is equivalent to } A, \\ 0 & \text{otherwise.} \end{cases} \tag{1}$$

Recent work has proposed various methods to improve LLM performance on the math problem solving task. In particular, majority voting, or self-consistency, can yield a significant improvement compared with the baseline performance of LLMs (Lewkowycz et al., 2022; Wang et al., 2023). Throughout this work, we will use the model's pass-at-1, pass-at-N, and majority voting performance for model evaluation and comparison. The specific definitions are:

(1) **Pass@1** (pass-at-1): the accuracy of the model's greedy-decoded solution $A_G$, i.e., $g(A, A_G)$.

(2) **Pass@N** (pass-at-N): the oracle performance that always selects the correct solution when it is presented in $N$ *temperature sampled* solutions, $\{\tilde{A}_1, \tilde{A}_2, ..., \tilde{A}_N\}$, i.e, $\max_{i \in \{1,2,...,N\}} g(A, \tilde{A}_i)$.

(3) **Maj1@N** (majority-voting-at-N): $N$ sampled solutions are first clustered by their math equivalence, i.e., $g(\tilde{A}_i, \tilde{A}_j)$. Then, one solution $\tilde{A}^*$ from a most frequent cluster is selected for calculating the accuracy, $g(A, \tilde{A}^*)$.

(4) **MajK@N**: Similar to Pass@N, we define an oracle that always selects the correct solution when it is presented in the top-K majority-voting clusters, $\{\tilde{A}_1^*, \tilde{A}_2^*, ..., \tilde{A}_K^*\}$, so its accuracy is $\max_{i \in \{1,2,...,K\}} g(A, \tilde{A}_i^*)$.

Another line of work leverages external tools such as Python programs to enhance the LLMs' ability (Chen et al., 2022; Wu et al., 2023; Yue et al., 2023; Zhou et al., 2023). In this work, we focus on improving the LLMs' inherent ability to solve math problems without help from external tools.

## 3 METHODS

### 3.1 SUPERVISED SOLUTION FINE-TUNING

In Hendrycks et al. (2021b); Cobbe et al. (2021) models are fine-tuned to generate not only the final answer but also the step-by-step process for solving the math problem.

$$S, A \leftarrow M(P), \tag{2}$$

where $P$ is the math problem, $S$, $A$ are the ground-truth step-by-step solution and the final answer respectively, and $M$ is an LLM. In training, the solution $S$ and the final answer $A$ are concatenated into a single text sequence $X$, and the model is fine-tuned with the cross-entropy loss following the maximum likelihood estimation (MLE) paradigm:

$$L_{\text{mle}} = -\log p_M(X|P), \tag{3}$$

where $p_M$ is the probably distribution given by the auto-regressive language model $M$:

$$p_M(X|P) = \prod_i p_M(x_i|X_{0,....,i-1}, P). \tag{4}$$

Here, $x_i$ is the $i$-th token in $X$, $X_{0,....,i-1}$ is the prefix before $x_i$.

To collect the ground-truth step-by-step solutions, we use two sources: (1) the original human-written solutions in the MATH dataset, (2) GPT-4 generated solutions provided in Lightman et al. (2023) with the chain-of-thought prompting eliciting step-by-step solutions. Our preliminary analysis found that the original solutions in the MATH dataset are more abstract while the solutions generated by GPT-4 are more fine-grained and detailed.

### 3.2 SOLUTION-CLUSTER RE-RANKING

We note that there are two significant gaps for LLMs' math problem solving performance in Table 2: (1) the gap between the model's greedy-decoding (Pass@1) and majority-voting (Maj1@N) results; (2) the gap between the model's majority-voting best-at-1 (Maj1@N) and best-at-K performance (MajK@N). To narrow these gaps, we fine-tune the pre-trained LLM as a solution verifier/evaluator, following Cobbe et al. (2021). However, unlike in the previous work where a large number (e.g., 1000) of candidate solutions are all reranked by the evaluator, we combine the strength of majority voting and re-ranking together by only re-ranking the top-K solution *clusters*. We believe this re-ranking strategy is both more robust and cost-efficient, as will be elaborated in the following section.

To use the evaluator to score each candidate solution, we formulate the scoring task as a classification problem in a text completion format, inspired by related work on using LLMs for text evaluation (Liu et al., 2023; Fu et al., 2023). Concretely, we define a mapping function $T$ converting the math problem $P$ and a candidate solution $\tilde{X}$ into a prompt $T(P, \tilde{X})$: "`Here is a math problem:` $P$. `Here is a candidate solution:` $\tilde{X}$. `The above candidate solution is` ". We then interpret the model-predicted probability of the word "correct" (or "incorrect") being the next token[1] as the probability of the solution being correct (or incorrect):

$$p_{\text{cls}}(\text{``correct''}|\tilde{X}, P) = p_M(\text{``correct''}|T(P, \tilde{X})), \tag{5}$$

$$p_{\text{cls}}(\text{``incorrect''}|\tilde{X}, P) = p_M(\text{``incorrect''}|T(P, \tilde{X})). \tag{6}$$

---

[1] We note that the tokenizer we used tokenizes the words "correct" and "incorrect" both into a single token.

We can then define the following normalized probability as the candidate solution score:

$$S_{\text{cls}}(\tilde{X}|P) = \frac{p_{\text{cls}}(\text{``correct''}|\tilde{X}, P)}{p_{\text{cls}}(\text{``correct''}|\tilde{X}, P) + p_{\text{cls}}(\text{``incorrect''}|\tilde{X}, P)}. \tag{7}$$

With this scoring format, we investigate two training objectives:

(1) **Margin loss for pairwise comparison**:

$$L_{\text{cls-margin}} = \max(0, \log S_{\text{cls}}(\tilde{X}_{\text{incorrect}}|P) - \log S_{\text{cls}}(\tilde{X}_{\text{correct}}|P) + \lambda), \tag{8}$$

where $\tilde{X}_{\text{correct}}$ and $\tilde{X}_{\text{incorrect}}$ stand for a correct and an incorrect solution respectively, and $\lambda$ is a hyper-parameter for the margin.

(2) **Cross-entropy loss for classification**. The scoring format we designed is equivalent to a multi-class classification problem where "correct" and "incorrect" are the only valid options. Therefore, we can fine-tune the model using the cross-entropy loss for this classification task:

$$\begin{aligned} L_{\text{cls-xent}} = &-\mathbb{1}_{\{\tilde{X}\text{ is correct}\}}(\tilde{X}) \log p_{cls}(\text{``correct''}|\tilde{X}, P) \\ &+ \mathbb{1}_{\{\tilde{X}\text{ is incorrect}\}}(\tilde{X}) \log p_{cls}(\text{``incorrect''}|\tilde{X}, P) \end{aligned} \tag{9}$$

### 3.3 MULTI-TASK SEQUENTIAL FINE-TUNING

The MLE-based training objective defined in Eq. 3 is somewhat at odds with the ultimate binary evaluation target – whether the final answer is correct or not. Related work has explored better aligning training with the task evaluation using a contrastive learning objective (Edunov et al., 2018; Liu et al., 2022; Zhao et al., 2023), which interprets the model-predicted probability of a candidate solution $\tilde{X}$ as its quality score and uses the margin loss to encourage the model to assign higher probabilities to better candidates:

$$L_{\text{seq}} = \max(0, \log p_M(\tilde{X}_{\text{incorrect}}|P) - \log p_M(\tilde{X}_{\text{correct}}|P) + \lambda). \tag{10}$$

Then, the model is fine-tuned with the MLE and contrastive training objectives jointly:

$$L_{\text{ctr}} = L_{\text{seq}} + \alpha_1 L_{\text{mle}}, \tag{11}$$

where $\alpha_1$ is a hyper-parameter.

However, the contrastive learning objective may not be as suitable for the math problem solving task because of the binary nature of the task – the objective requires the model to use the token likelihoods for two purposes: (1) predicting the next token, and (2) evaluating the quality of the entire text sequence. It can be a reasonable objective for natural language generation tasks such as text summarization where the next token prediction correctness is closely related to overall text quality. However, for math problem solving, the correctness of a solution might be decided by just a few tokens, making the next-token prediction task more distant and incompatible with the solution evaluation task. Consequently, we combine the training objectives in §3.1 (Eq. 3) and §3.2 (Eq. 8 and Eq. 9), introducing a new learning setting that formulates both math problem solution generation and evaluation tasks as natural language generation tasks:

$$L_{\text{mul-margin}} = L_{\text{cls-margin}} + \alpha_2 L_{\text{mle}}, \tag{12}$$

$$L_{\text{mul-xent}} = L_{\text{cls-xent}} + \alpha_3 L_{\text{mle}}, \tag{13}$$

where $\alpha_2$ and $\alpha_3$ are hyper-parameters. We believe we can better leverage the capacity of LLMs with this training setting since it is closer to the pre-training task (i.e., next-token prediction). In our preliminary experiments, we found that it is difficult to balance the two loss terms in Eq. 12 and Eq. 13 and the models start to overfit the MLE training objective very soon, possibly because of the limited size of the dataset. As the result, we optimize the multi-task objective in a *sequential* manner – instead of fine-tuning the model on both training objectives, we first fine-tuned the model as a generator (Eq. 8 or Eq. 9), then as an evaluator (Eq. 3), then finally as a generator again.

Table 1: Number of examples in dataset splits and the average length in tokens of the math problems and solutions.

| Data Source | # Training | # Validation | # Test | Problem Length | Solution Length |
|---|---|---|---|---|---|
| MATH | 11000 | 1000 | 500 | 90.2 | 249.6 |
| PRM800K | 6473 | 564 | 512 | 57.3 | 305.2 |

Table 2: Results of supervised solution fine-tuning. Different training data sources are compared, which are the MATH dataset and the PRM800K dataset.

| | PaLM 2-S* | | | PaLM 2-L | | |
|---|---|---|---|---|---|---|
| | Pass@1 | Maj1@64 | Pass@64 | Pass@1 | Maj1@64 | Pass@64 |
| Few-shot | 17.4% | 27.2% | 67.8% | 33.4% | 47.6% | 79.4% |
| MATH | 19.8% | 32.4% | 74.8% | 32.8% | 49.2% | 82.2% |
| PRM800K | 20.8% | 41.2% | 73.8% | 34.8% | 54.2% | 83.4% |
| MATH + PRM800K | 22.6% | 38.8% | 72.6% | 35.6% | 55.2% | 82.8% |

## 4 EXPERIMENTS

### 4.1 EXPERIMENTAL SETTING

**Datasets** Our experiments are conducted on the MATH dataset. To prevent overfitting, we follow Lightman et al. (2023) by using the data splits they provided,[2] where 4.5K original test examples are used for training and the remaining 500 test examples are used for model evaluation. We leverage two sources of correct step-by-step solutions for the model training: (1) the original human-written explanations provided in the MATH dataset; (2) the model-generated correct solutions provided in PRM800K (Lightman et al., 2023), which only covers a subset problems in the original MATH dataset. The dataset statistics are provided in Table 1.

**Evaluation** We report the average solution accuracy (or correctness) for all the experiments. The correctness of the generated solution is compared against the ground-truth solution using the automatic grading script provided by Lightman et al. (2023). This script checks the mathematical equivalence instead of the simple textual equivalence. Two solution generation methods are mainly used to evaluate the model performance: (1) greedy decoding for the Pass@1 performance, (2) nucleus sampling (Holtzman et al., 2020) for the majority voting performance (Maj1@N), where we used the same sampling hyper-parameters as in Lewkowycz et al. (2022). Specifically, the sampling temperature is set to 0.6, and the top-$p$ value is set to 0.95.

### 4.2 EXPERIMENT I: SUPERVISED SOLUTION FINE-TUNING

We fine-tune PaLM 2-S* and PaLM 2-L on the step-by-step solutions with the MLE training objective (Eq. 3). Three specific fine-tuning strategies are explored: (1) fine-tuning using the original MATH solutions only; (2) fine-tuning using the PRM800K GPT-4 solutions only; (3) fine-tuning on both MATH and PRM800K solutions. We used the model performance on the validation set for checkpoint selection, and all the fine-tuned models achieved the best performance within two epochs. The results are shown in Table 2, where the few-shot performance with the pre-trained PaLM 2 models are provided for comparison. The few-shot results are obtained using a customized 4-shot prompt designed in Lewkowycz et al. (2022).

We observe that the fine-tuning is generally helpful for the model to achieve better performance compared with the few-shot performance of the pre-trained checkpoints. Moreover, the quality and the style of the solutions can have a large impact on the model performance, since the models fine-tuned on PRM800K solutions achieve significantly better performance than the ones fine-tuned on the original MATH solutions.

---

[2] https://github.com/openai/prm800k

Table 3: Results of solution-cluster re-ranking. Two loss functions are compared, i.e., $L_{\text{cls-margin}}$ (Eq. 8) and $L_{\text{cls-xent}}$ (Eq. 9). Two re-ranking strategies are used: (1) re-ranking all the candidate solutions (RR.All), (2) re-ranking all solutions in the top-8 solution clusters (RR.Top-8). The baseline performance (Pass@1 and Maj1@64) are reported for comparison.

| Model | Loss Function | Pass@1 | Maj1@64 | RR.All | RR.Top-8 |
|-------|---------------|--------|---------|--------|----------|
| PaLM 2-S* | $L_{\text{cls-margin}}$ | 22.6% | 38.8% | 32.4% | 36.6% |
|           | $L_{\text{cls-xent}}$ | 22.6% | 38.8% | 33.6% | 35.4% |
| PaLM 2-L | $L_{\text{cls-margin}}$ | 35.6% | 55.2% | 57.0% | 58.8% |
|          | $L_{\text{cls-xent}}$ | 35.6% | 55.2% | 56.8% | 58.8% |

Table 4: Results of multi-task sequential fine-tuning. The model performance of the generator fine-tuned with the multi-task sequential setting is compared with the baseline generator trained with the MLE training objective only. Two model variants with different solution evaluation training objectives ($L_{\text{cls-margin}}$ and $L_{\text{cls-xent}}$) are compared. All the model checkpoints are PaLM 2-L based.

| Baseline | Evaluator Loss Function | Pass@1 | Maj1@64 | Pass@64 |
|----------|-------------------------|--------|---------|---------|
| Yes | - | 35.6% | 55.2% | 82.8% |
| No | $L_{\text{cls-margin}}$ | 37.6% | 57.2% | 82.6% |
| No | $L_{\text{cls-xent}}$ | 36.2% | 56.6% | 82.2% |

### 4.3 EXPERIMENT II: SOLUTION-CLUSTER RE-RANKING

In Table 2, we observe that there is a large performance gap between the model's Pass@1 and Pass@64 performance, indicating that the model has already possessed the ability to search for correct solutions, but fails to differentiate its different search results. Therefore, we continue fine-tuning the models as *solution evaluators* for the solution re-ranking task.

We investigate two loss functions from §3.2, the margin loss (Eq. 8) and the cross-entropy loss (Eq. 9) for the model training. For Eq. 8 we found that the model performance is not sensitive to the margin hyper-parameter $\lambda$, so we keep it fixed as $\log 2$ which intuitively requires the model to assign at least twice the probability to the correct solution. The checkpoints from §3.2 that are trained on both MATH and PRM800K solutions are used for this experiment, and the 64 candidate solutions are sampled from these checkpoints themselves for each problem. They are used to construct 10 training examples for the margin loss, and all of them are used for the cross-entropy loss. We observe all of the experiments converged within one epoch, possibly because of the redundancy in the training data. We make the following observations based on the results in Table 3:

(1) For both PaLM 2-S* and PaLM 2-L, the re-ranking result can outperform the Pass@1 performance of the baseline model. However, only the PaLM 2-L evaluator can outperform the robust majority-voting baseline, showing the re-ranking is a difficult task for relatively smaller models.

(2) Re-ranking only the solutions in the top solution clusters are consistently better than re-ranking all the solutions for both PaLM 2-S and PaLM 2-L. This re-ranking strategy is also more computationally efficient since there are fewer solutions to rank.

We provide further analysis in §5 and Appendix B.

### 4.4 EXPERIMENT III: MULTI-TASK SEQUENTIAL FINE-TUNING

Having shown that better performance can be gained by fine-tuning the LLMs as a solution evaluator, we now investigate whether the training objective for solution evaluation is also helpful for the models to become better solution *generators*. To this end, we aim to use the multi-task sequential training objectives (Eq. 12 and Eq. 13) we proposed in §3.3 for the model fine-tuning: (1) the first step is exactly the experiments in §4.2, where the model is fine-tuned as a *solution generator*; (2) the model is then fine-tuned as a *solution evaluator* as in §4.3; (3) the evaluator is fine-tuned again with the MLE training objective to regain its ability as a *solution generator*. We note that in the last step the models are only trained for around 200 steps before they achieve the best performance.

Table 5: Analysis of the effect of the reference solution style and quality on the few-shot performance of pre-trained models and zero-shot performance of fine-tuned models.

|  | PaLM 2-S* | | | PaLM 2-L | | |
|---|---|---|---|---|---|---|
|  | Pass@1 | Maj1@64 | Pass@64 | Pass@1 | Maj1@64 | Pass@64 |
| MATH Few-shot | 17.4% | 27.2% | 67.8% | 33.4% | 47.6% | 79.4% |
| PRM800K Few-shot | 19.2% | 27.8% | 70.0% | 31.4% | 47.2% | 82.4% |
| MATH Fine-tuned | 19.8% | 32.4% | 74.8% | 32.8% | 49.2% | 82.2% |
| PRM800K Fine-tuned | 20.8% | 41.2% | 73.8% | 34.8% | 54.2% | 83.4% |

Table 6: Generalization ability of the solution evaluator. The evaluators are tested on solutions generated from different models. The evaluator checkpoints are trained with $L_{\text{cls-xent}}$ (Eq. 9).

| Evaluator | Solution Generator | Pass@1 | Maj1@64 | RR.All | RR.Top-8 |
|---|---|---|---|---|---|
| PaLM 2-S* | PaLM 2-S* | 22.6% | 38.8% | 32.4% | 36.6% |
|  | PaLM 2-L | 35.6% | 55.2% | 48.4% | 50.6% |
| PaLM 2-L | PaLM 2-S* | 22.6% | 38.8% | 46.0% | 46.4% |
|  | PaLM 2-L | 35.6% | 55.2% | 56.8% | 58.8% |

The results in Table 4 show that the models fine-tuned with the multi-task learning objective can achieve better performance than models fine-tuned on the MLE training objective only (§4.2). It indicates that the training objective of the solution *evaluation* task can provide useful supervision signals to the solution *generation* model. We believe this is because formulating the solution evaluation task as next-word prediction can better leverage the LLM's ability gained during pre-training.

## 5   ANALYSIS

### 5.1   UNDERSTANDING THE EFFECT OF STEP-BY-STEP SOLUTION STYLE

In §4.2, we found that the models fine-tuned on the GSM800K solutions significantly outperform the ones fine-tuned on the original MATH solutions. We hypothesize this is because the GSM800K solutions are finer-grained and follows more closely to the step-by-step solution format (an example is shown in Figure 2). To investigate whether this difference can also affect the pre-trained models' few-shot performance, we rewrite the custom 4-shot prompt used in Lewkowycz et al. (2022) by replacing the original MATH solution in the prompt with the GSM800K solution. The GSM800K solution is missing for one problem, which we replace with a similar problem with a valid solution.

Table 5 shows that the few-shot performance is relatively invariant to the difference of the reference solutions, indicating that fine-tuning is necessary for the model to benefit from the potentially higher-quality solutions. We leave it as future work to investigate whether fine-tuning on the model's own generated solutions with the same style can achieve a similar effect (Zelikman et al., 2022).

### 5.2   EVALUATING THE GENERALIZATION ABILITY OF THE SOLUTION EVALUATOR

In §4.3, the PaLM 2-L solution evaluator shows a strong performance at re-ranking the candidate solutions generated by the related PaLM 2-L fine-tuned solution generator. We now investigate whether this evaluator can also be used to re-rank solutions generated by other models. The results in Table 6 prove the generalization ability of the fine-tuned PaLM 2-L evaluator. On the other hand, PaLM 2-S* is ineffective at re-ranking both the PaLM 2-L and PaLM 2-S* solutions, which suggests that solution evaluation is a non-trivial task that requires sufficiently large models.

### 5.3   COMPARING DIFFERENT SOLUTION RE-RANKING STRATEGIES

Previous work has proposed various re-ranking strategies for math problem candidate solutions. Therefore, we compare them using the fine-tuned PaLM 2-L evaluators with respect to their performance in re-ranked solution accuracy and efficiency. The re-ranking strategies compared are:

Table 7: Comparison of different re-ranking strategies with the PaLM 2-L evaluator fine-tuned with $L_{\text{cls-margin}}$ (Eq. 8) and $L_{\text{cls-xent}}$ (Eq. 9) respectively. The optimistic performance with the optimal hyper-parameter configuration is reported.

| Loss Function | RR.All | RR.MajK | W.RR | W.RR.MajK | Maj1 | Maj1.TopN |
|---|---|---|---|---|---|---|
| $L_{\text{cls-margin}}$ | 57.0% | 59.4% | 60.8% | 60.8% | 55.2% | 59.4% |
| $L_{\text{cls-xent}}$ | 56.8% | 59.4% | 60.8% | 61.2% | 55.2% | 60.0% |

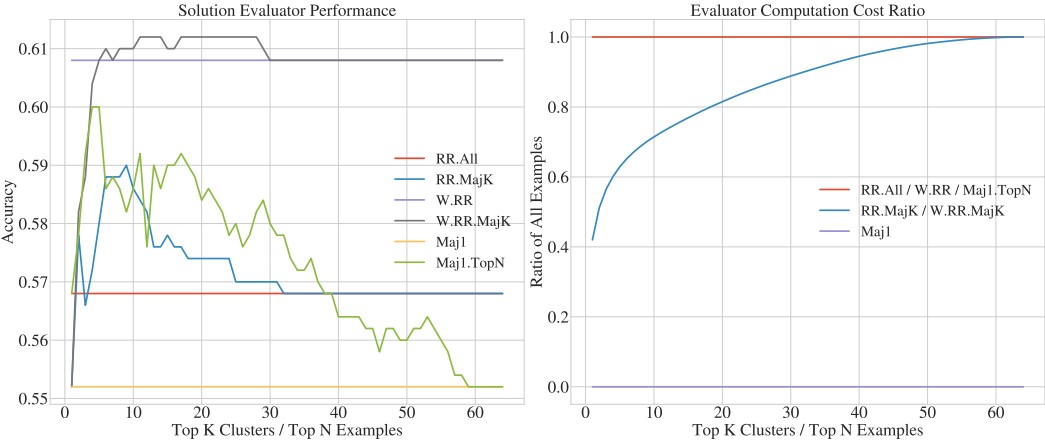

Figure 1: Analysis of different re-ranking strategies. In the left figure, the performance of different re-ranking methods are plotted against the hyper-parameters. In the right figure, the evaluator computation cost is computed. The PaLM 2-L evaluator trained with the $L_{\text{cls-xent}}$ (Eq. 9) is used.

1. Vanilla re-ranking (**RR.All**). The final solution $\tilde{X}^*$ is selected via

$$\tilde{X}^* = \arg\max_{\tilde{X} \in \tilde{\mathcal{X}}} S_{cls}(\tilde{X}|P), \tag{14}$$

where $S_{cls}$ is the scoring function parameterized by the solution evaluator (Eq. 7), $P$ is a math problem, $\tilde{\mathcal{X}}$ is a set of candidate solutions.

2. MajK re-ranking (**RR.MajK**). This is the strategy we used in §4.3, which only re-ranks the set of solutions $\tilde{\mathcal{X}}_K$ in the top-K clusters from majority voting:

$$\tilde{X}^* = \arg\max_{\tilde{X} \in \tilde{\mathcal{X}}_K} S_{cls}(\tilde{X}|P). \tag{15}$$

3. Weighted re-ranking (**W.RR**). This strategy is proposed in Li et al. (2023) and adopted by Uesato et al. (2022), re-ranking answer clusters according to the sum of the answer scores in each clusters:

$$\tilde{X}^* = \arg\max_{\tilde{X} \in \tilde{\mathcal{X}}} \sum_{\hat{X} \in \mathcal{X}} g(\tilde{X}, \hat{X}) S_{cls}(\hat{X}|P), \tag{16}$$

where $g$ is the auto-grader that assigns 1 when two answers are equivalent and 0 otherwise (Eq. 1).

4. Weighted MajK re-ranking (**W.RR.MajK**). It combines re-ranking strategy 2 and 3 together:

$$\tilde{X}^* = \arg\max_{\tilde{X} \in \tilde{\mathcal{X}}_K} \sum_{\hat{X} \in \mathcal{X}_K} g(\tilde{X}, \hat{X}) S_{cls}(\hat{X}|P). \tag{17}$$

5. (Self-consistency) majority voting (**Maj1**) (Wang et al., 2023):

$$\tilde{X}^* = \arg\max_{\tilde{X} \in \tilde{\mathcal{X}}} \sum_{\hat{X} \in \tilde{\mathcal{X}}} g(\tilde{X}, \hat{X}). \tag{18}$$

6. Majority voting of top-N solutions (**Maj1.TopN**). Cobbe et al. (2021) proposes a method that applies majority voting only on the top-N solutions $\tilde{\mathcal{X}}_N$ selected by the solution evaluator:

$$\tilde{X}^* = \arg \max_{\tilde{X} \in \tilde{\mathcal{X}}_N} \sum_{\hat{X} \in \tilde{\mathcal{X}}_N} g(\tilde{X}, \hat{X}). \tag{19}$$

In Table 7, we show the *optimistic* performance of each re-ranking strategy with the best hyper-parameter configuration using the PaLM 2-L re-rankers and 64 generated candidate solutions. In Figure 1 we provide a detailed analysis with respect to both the re-ranking performance (in the left figure) and the computation efficiency (in the right figure). We found that the weighted re-ranking strategy (W.RR) has storng performance, and the modified version we proposed (W.RR.MajK) can achieve comparable performance while reducing the computational cost of the solution evaluator.

# 6 RELATED WORK

Hendrycks et al. (2021b) introduced the MATH dataset and fine-tuned GPT-2 (Radford et al., 2019) and smaller GPT-3 language models using (1) solution-and-answer and (2) answer-only as targets, but achieving less than 7% test accuracy, demonstrating the difficulty of the task at the time. Subsequently Cobbe et al. (2021) introduced GSM8K, a similar but easier, elementary-school math problem-solving dataset and showed that sampling followed by re-ranking using a trained verifier (or reward model) could improve performance significantly over a single sample.

Recent work has explored different methods for improving the LLMs' math solving ability. Specifically, Lewkowycz et al. (2022) proposes to continue pre-training the LLMs on math-specific corpora, gaining a significant improvement from the original pre-trained models. Uesato et al. (2022) extends outcome verifiers/reward-models (ORMs) to process reward models (PRMs) to judge solutions at the step-level by collecting human-annotated labels. They report a negative result in reranking using PRMs compared to ORMs on GSM8K but use them successfully to tune models using reinforcement learning. Lightman et al. (2023) in contrast scaled human annotation of process-level labels dramatically and achieved improved performance from reranking using PRMs on MATH with GPT-4 (OpenAI, 2023) as the base model achieving state-of-the-art performance.

Apart from training methods that directly improve the base model performance, inference-time, prompt-engineering techniques, such as chain-of-thought (Nye et al., 2021; Wei et al., 2022; Kojima et al., 2022) and self-consistency majority voting (Wang et al., 2023), have also demonstrated their effectiveness when using strong base large language models such as PaLM/PaLM2 (Chowdhery et al., 2022; Anil et al., 2023) and GPT-3/4 (Brown et al., 2020; OpenAI, 2023), to the point that state-of-the-art performance on GSM8K is nearing 100%, and hence renewed focus on the harder MATH task. Some recent work (Chen et al., 2022; Wu et al., 2023; Yue et al., 2023) focuses on leveraging external tools, such as Python programs, to complement the LLM's ability, which shows further improvement over pure LLM-based methods. However we focus on the setting with no tools.

Zelikman et al. (2022) shows that iteratively sampling and fine-tuning to the correct model solutions can improve mathematical performance (GSM8K). Although we do not employ this technique, it is orthogonal and conceivably may further improve performance.

# 7 CONCLUSION

In this work we investigated different fine-tuning methods to improve the LLMs' performance on math problem solving. Starting with supervised step-by-step fine-tuning, we first demonstrated the importance of step-by-step solutions for improving fine-tuned LLM performance. We then studied re-ranking methods for fine-tuning the LLMs as solution evaluators, and proposed a new re-ranking method which combines the benefit of majority voting and re-ranking together, simultaneously achieving better solution accuracy and computational efficiency. Lastly, we introduced a multi-task sequential fine-tuning method, aiming at improving the model's solution generation ability with the training objective of the solution evaluation. Our method outperforms the baseline fine-tuning method based on the solution generation training objective only, demonstrating its ability of improving a generation task using the supervision signal of the corresponding evaluation task.

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

# A  EXAMPLES OF STEP-BY-STEP SOLUTIONS

---

**Problem**

What is the greatest common factor of 20! and 200,000? (Reminder: If $n$ is a positive integer, then $n!$ stands for the product $1 \cdot 2 \cdot 3 \cdots \cdots (n-1) \cdot n$.)

**MATH Solution**

The prime factorization of $200,000$ is $2^6 \cdot 5^5$. Then count the number of factors of 2 and 5 in 20!. Since there are 10 even numbers, there are more than 6 factors of 2. There are 4 factors of 5. So the greatest common factor is $2^6 \cdot 5^4 = \boxed{40,000}$.

**PRM800k solution**

I want to find the largest positive integer that divides both 20! and 200,000 evenly. One way to do this is to factor both numbers into prime factors and look for the common ones.
I know that $200,000 = 2^5 \cdot 10^4 = 2^9 \cdot 5^4$.
To find the prime factorization of 20!, I can use the fact that it is the product of all the positive integers from 1 to 20.
For each prime number $p$ between 1 and 20, I can count how many multiples of $p$ are in that range.
For example, there are 10 multiples of 2 between 1 and 20, namely $2, 4, 6, \ldots, 20$.
But there are also 5 multiples of 4, which is $2^2$, and 2 multiples of 8, which is $2^3$, and 1 multiple of 16, which is $2^4$.
So, the total power of 2 in 20! is $10 + 5 + 2 + 1 = 18$.
Similarly, there are 4 multiples of 5, namely $5, 10, 15, 20$, so the power of 5 in 20! is 4.
There are 6 multiples of 3, namely $3, 6, 9, \ldots, 18$, but there are also 2 multiples of 9, which is $3^2$, so the power of 3 in 20! is $6 + 2 = 8$.
There are 2 multiples of 7, namely 7 and 14, so the power of 7 in 20! is 2.
There are 1 multiple of each of the other prime numbers $11, 13, 17,$ and $19$, so the powers of those primes in 20! are 1 each.
Therefore, the prime factorization of 20! is $2^{18} \cdot 3^8 \cdot 5^4 \cdot 7^2 \cdot 11 \cdot 13 \cdot 17 \cdot 19$.
To find the greatest common factor of 20! and 200,000, I need to take the lowest power of each common prime factor.
The only common prime factors are 2 and 5, and the lowest powers are 9 and 4, respectively.
So, the greatest common factor is $2^9 \cdot 5^4 = 512 \cdot 625 = 320,000$.

# Answer

320,000

---

Figure 2: Example comparing MATH and GPT-4 generated step-by-step solutions.

# B  ANALYSIS OF SOLUTION CLUSTERS

We conduct an in-depth analysis to better understand the property of the solution clusters. We first analyze the performance upper-bound of the solution-cluster re-ranking approach in Figure 3, which indicates that the evaluators we trained in §4.3 can still be further improved. In fact, with a perfect solution evaluator, an accuracy of 64.0% can be achieved by re-ranking just the top-2 clusters, while the majority-voting performance is only 55.2%.

In Figure 4, we further investigate the difficulty of the re-ranking task by analyzing the characteristics of the PaLM 2-L solution clusters. Specifically, we compare the size of the correct solution cluster against (1) the number of all sampled solutions, (2) the size of the top-1 solution cluster when the correct solution is not selected by the majority voting. The results show two trends: (1) in general the re-ranking task is difficult, since the correct solutions only consist of less than 5% of all solutions in 50% of the time; (2) re-ranking top-clusters is relatively easier, since the ratio of the size of correct solution cluster and the top-1 solution cluster is much more balanced. We believe these observations can partially explain the benefit of the solution-cluster re-ranking method we proposed.

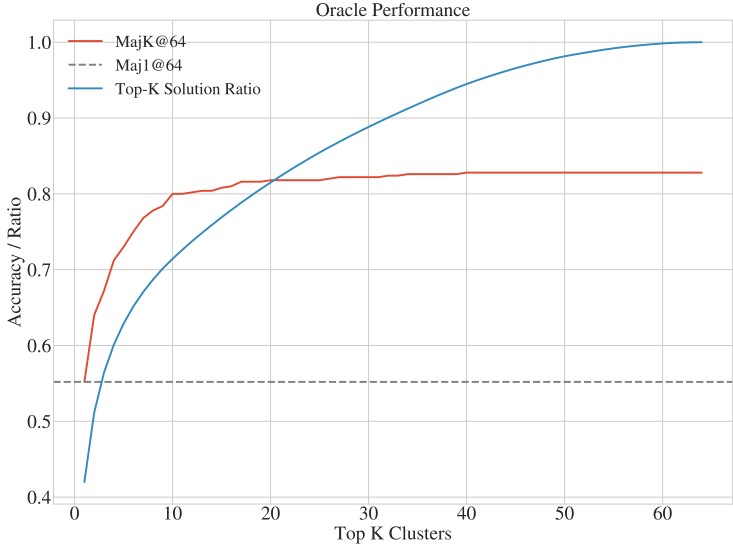

Figure 3: Performance comparison of the majority voting (Maj1@64) and the oracle that always selects the correct solution in the top-K clusters (MajK@64).

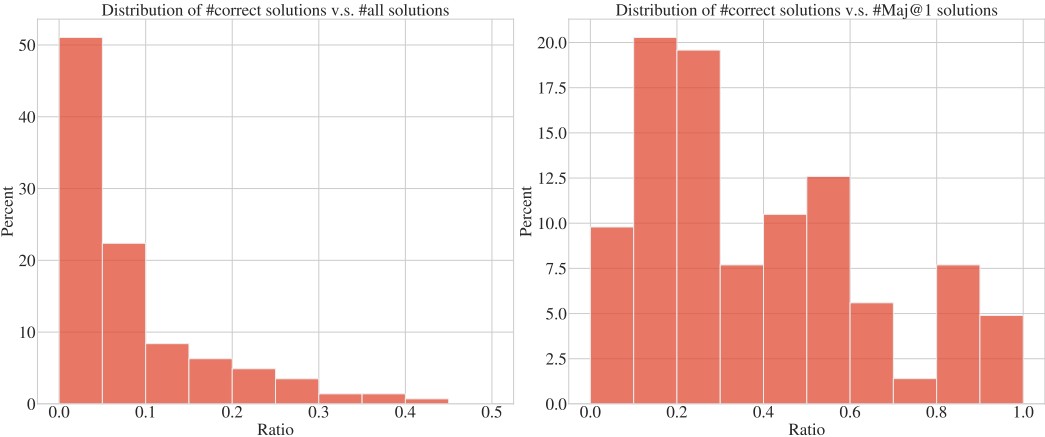

Figure 4: Distributions of (1) the ratio of the number of correct solutions v.s. all solutions, (2) the ratio of number of correct solutions v.s. Maj1@64 solutions when the correct solution is not the Maj1@64 solution.

