# OpenReview forum: "Improving Large Language Model Fine-tuning for Solving Math Problems"
_ICLR.cc/2024/Conference — Submitted to ICLR 2024_

### Official Review · Reviewer_1Ei9 · 2023-11-02

**Soundness:** 2 fair
**Presentation:** 2 fair
**Contribution:** 2 fair
**Rating:** 3
**Confidence:** 4

**Summary:**

The paper explores methods to improve large language models (LLMs) for solving math word problems.
The authors identify a gap between LLMs' pass-at-one (single attempt) and pass-at-N (multiple attempts) performance on math problems, suggesting LLMs can often find correct solutions but struggle to identify them.
Three fine-tuning strategies are proposed to improve LLMs' solution generation and evaluation:

- Supervised fine-tuning to generate step-by-step solutions
- Solution-cluster re-ranking, where the LLM ranks candidate solutions clustered by equivalence
- Multi-task sequential fine-tuning, combining generation and evaluation

Experiments on the MATH dataset with PaLM 2 models show the benefits of each strategy.

**Strengths:**

- The solution-cluster re-ranking approach creatively combines majority voting and re-ranking in a novel way to improve performance.
- The paper provides clear motivation,

**Weaknesses:**

- Lack of comparison with other open-source LLMs.
- Lack of comparison with other powerful LLMs such as ChatGPT [1], GPT-4 [2], and Claude-2 [3].
- The experiments are solely conducted on PaLM 2 models, not demonstrating generalizability to other model families such as LLaMA [4].
- Compared to the baseline, the performance gain is minimal.

[1] OpenAI. (2022). Introducing chatgpt. https://openai.com/blog/chatgpt, 2022.
[2] OpenAI (2023). GPT-4 Technical Report
[3] Anthropic (2022). Instroducing claude. https://www.anthropic.com/index/introducing-claude
[4] Touvron et al (2023). LLaMA: Open and Efficient Foundation Language Models

**Questions:**

Can you show the experiment results on LLaMA, LLaMA-2[5] on 7B/13B?
Can you compare your results with ChatGPT, GPT-4  and Claude-2?

[5] Touvron et al (2023). Llama 2: Open Foundation and Fine-Tuned Chat Models

---

### Official Review · Reviewer_ShdB · 2023-11-05

**Soundness:** 2 fair
**Presentation:** 2 fair
**Contribution:** 1 poor
**Rating:** 3
**Confidence:** 4

**Summary:**

This paper experiments several fine-tuning strategies to improve the LLM performance on math word problems. They also proposed
1.  _Solution-cluster re-ranking_, which reduces the load of re-ranking a large number of candidates by just re-ranking top-_K_ clusters;
2.  _Multi-task sequential fine-tuning_: sequentially fine-tune the model as a generator, then evaluator, and then generator again.

**Strengths:**

1. Experiments (Table 3) show consistent improvement by __RR.Top-8__ compared with __RR.All__.
2. Improvement of the sequential fine-tuning over the MLE fine-tuning.
3. Investigate a number of re-ranking strategy to show the advantages of reranking through top-_K_ clusters.

**Weaknesses:**

1. The contribution is incremental to the research community.
    1. We can see the absolute improvements are not really significant for both solution reranking and sequential fine-tuning (Table 3 and 4).
    2. Thus, it could be unnecessary to make this approach general for everyone.
    3. I would expect some deeper insights besides the engineering efforts made in this work. For example, how does the sequential fine-tuning change the behavior of the models? Answering some scientific questions like this would give the readers more insights for future research.
2. Insufficient and unclear experiments.
    1. Reranking: why K is 8? I think there should be more experiments to explain.
    2. One more dataset would be even more convincing.
3. Writing needs to be improved
     1. For example, in Section 3.2, we are following Cobbe et al., (2021) to perform re-ranking. When you have the pair loss in (7-9), it is obvious that we don't explicitly have the correct and incorrect pairs. The number sampled from the model could be unbalanced, how do you handle this problem. And, do you still use the gold annotations in this reranking?

**Questions:**

1. is equation 10 and 11 used in the experiments? Seems only (12) and (13) are used. Seems these two equations are useless there?

---

> ### Author Response · Authors · 2023-11-22
>
> > Reranking: why K is 8? I think there should be more experiments to explain.
>
> Thank you for the suggestion. In the left subfigure of Figure 1, we illustrate the effect of K. We observed that the improvement in solution-cluster re-ranking remains consistent across a wide range of K values.
>
> > Writing needs to be improved. For example, in Section 3.2, ... When you have the pair loss in (7-9), it is obvious that we don't explicitly have the correct and incorrect pairs. The number sampled from the model could be unbalanced, how do you handle this problem. And, do you still use the gold annotations in this reranking?
>
> In Section 4.3, we describe our data construction process. We used the model, trained with supervised solution fine-tuning, to generate 64 candidate solutions for each data example. Utilizing these solutions and the auto-grader, we constructed 10 correct-incorrect solution pairs, without relying on gold annotations.
>
> > is equation 10 and 11 used in the experiments?
>
> We did not incorporate them in the experiments; they are introduced solely for discussion purposes.

---

### Official Review · Reviewer_hJqw · 2023-11-09

**Soundness:** 2 fair
**Presentation:** 2 fair
**Contribution:** 2 fair
**Rating:** 3
**Confidence:** 2

**Summary:**

This work aims to improve the LLMs’ performance on math problem solving. They adopt several methods. (1) Using step-by-step fine-grained solutions. (2) Using majority voting with re-ranking method to choose final solution. (3) multi-tasks tuning: solution generation task and solution evaluation task.

**Strengths:**

(1) Using reranking+ majority voting to choose solution and compare different re-ranking algorithm, different model size.
(2) This work propose the multi-tasks tuning in a sequential manner while a new training object.

**Weaknesses:**

(1) I find its novelty is limited. Both reranking and majority-voting are not new, it seems the authors just combine them in this paper.
(2) Multi-task sequential fine-tuning is one of major contribution. However, I find it increase performance very limited as shown in Table 4.
(3) The paper claims the method obtains an 11.2% accuracy improvement. However, the simply SFT on PRM800+MATHalready has about 7.6% gain as shown in Table2  (47.6->55.2) , so the biggest gain is from SFT instead of proposed method.
(4) The paper only reports the performance on PaLM2. I think the authors can report the method on other LLM, like llama.
(5) In sec 5.1, I am not sure what's the dataset GSM800K? maybe it is PRM800K. The authors should carefully check the paper writing.

**Questions:**

(1) The authors train the model by a sequential manner, I wonder what's the result if the three tasks are trained together. Or what's the result if only two tasks are used.

(2) I am not sure the authors combine the reranking in the Section 4.4 or not. If it is not, what about the performance of reranking + majority-voting + multi-task tuning?

---

> ### Author Response · Authors · 2023-11-22
>
> > In sec 5.1, I am not sure what's the dataset GSM800K? maybe it is PRM800K.
>
> Thank you for spotting this typo. Yes it should be PRM800K.
>
> > The authors train the model by a sequential manner, I wonder what's the result if the three tasks are trained together. Or what's the result if only two tasks are used.
>
> There are two tasks involved: (1) solution re-ranking and (2) solution generation. We observed that training the models together leads to easy overfitting on the second task.
>
> > I am not sure the authors combine the reranking in the Section 4.4 or not. If it is not, what about the performance of reranking + majority-voting + multi-task tuning?
>
> In Section 4.4 we train the model in a sequential manner, i.e., (1) training the model as a solution generator, (2) training the model as a solution evaluator (re-ranker), (3) training the model again as a solution generator. We report the performance of the model as a generator.

---

### Author Response · Authors · 2023-11-22
**General Response**

We thank the reviewers for their constructive comments. Due to time constraints, we are unable to complete the additional experiments requested at this moment and substantially improve the current submission. Therefore, we fully understand the reviewers' concerns and would respect any decisions made.

---

### Meta-Review · Area_Chair_bjF4 · 2023-12-07

**Metareview:**

This paper presents several methods for improving the LLMs’ performance on math problem solving, including  solution fine-tuning, solution-cluster re-ranking, and multi-task sequential fine-tuning. An empirical study is conducted on PaLM 2 models and the results are presented and discussed. The motivation of the work is clear and the proposed methods sound reasonable.

However, the novelty of the proposed methods is limited and the evaluation is not sufficient. It would be more convincing to use more datasets for evaluation and compare with other LLMs. The experiments are solely conducted on PaLM 2 models, not demonstrating generalizability to other model families such as LLaMA.

**Justification For Why Not Higher Score:**

see the meta-review.

**Justification For Why Not Lower Score:**

N/A

---

### Decision · Program_Chairs · 2024-01-16

Reject